# Insights into the Volatile Flavor Profiles of Two Types of Beef Tallow via Electronic Nose and Gas Chromatography–Ion Mobility Spectrometry Analysis

**DOI:** 10.3390/foods13101489

**Published:** 2024-05-11

**Authors:** Ke Li, Liangyao Zhang, Danhui Yi, Yunxiao Luo, Chao Zheng, Yinglong Wu

**Affiliations:** 1College of Food Science, Sichuan Agricultural University, Yaan 625014, China; like2341@126.com (K.L.); 13541228898@163.com (L.Z.); yidanhui53@gmail.com (D.Y.); rorin66@163.com (Y.L.); zc1952@163.com (C.Z.); 2Institute of Agriculture Products Processing Science and Technology, Sichuan Academy of Agriculture Science, Chengdu 610039, China

**Keywords:** beef tallow, E-nose, GC-IMS, OPLS-DA, flavor profile

## Abstract

In the current study, an electronic nose (E-nose) and gas chromatography–ion mobility spectrometry (GC-IMS) were employed to investigate the volatile flavor compounds (VFCs) of intense flavor beef tallow (L) and ordinary beef tallow (P). The study results indicate that an E-nose combined with an LDA and GC-IMS combined with an OPLS-DA can effectively distinguish between the two types of beef tallow. Compared with ordinary beef tallow, the E-nose sensors of intense flavor beef tallow have stronger response signals to sulfides, terpenes, and nitrogen oxides. A total of 22 compounds contribute to making the flavor of intense flavor beef tallow more typical and richer; in contrast, ethyl acetate was the main aroma-active compound found in the ordinary beef tallow. Sulfur-containing compounds and terpenoids might be the key substances that cause sensory flavor differences between the two types of beef tallow. In conclusion, the results of this study clarify the characteristics and differences of the two types of beef tallow and provide an enhanced understanding of the differences in the flavors of the two types of beef tallow.

## 1. Introduction

Refined fat, also known as beef tallow, is a type of edible animal fat made by removing impurities through a series of complex processes such as high-temperature refinement, degumming, deacidification, decolorization, and deodorization [1,2,3]. In addition to its desirable physical properties, such as its good heat stability, strong antioxidant properties, and mild taste with a pure and authentic flavor, beef tallow contains a variety of fatty acids, vitamins, and minerals [4]. Due to these properties, it is widely used in margarine, condiments, artificial animal creams, plasticized oils, shortening, hotpots, etc.

In hotpot seasoning and condiments, beef tallow is the main raw material that is used [4]. Its special animal fat aroma can better cover the unpleasant flavors of various ingredients, and its suitable solubility helps it to absorb the spicy, pungent, and numbing flavors of peppers and peppercorns [5]. This helps to make hot pots not only spicy, but also mild and safe, resulting in a red-colored soup with a rich aroma and rich taste without being greasy. Therefore, the quality of the hotpot condiment is fundamentally determined by the quality of the beef tallow. Regarding spicy hotpot products from Chongqing and Sichuan, China, beef tallow is an irreplaceable ingredient in hotpot seasoning and has become one of the most distinctive dietary ingredients in Chongqing and Sichuan [6].

The beef tallow used for hotpot on the market can be divided into two types: intense flavor beef tallow (Lao huo guo beef tallow) and ordinary beef tallow (Putong beef tallow). The two types of beef tallow differ in terms of raw materials, refinement processes, refinement degrees, and product quality characteristics. The raw materials used for intense flavor beef tallow include grass-fed beef, which is mainly derived from Xinjiang and Inner Mongolia, and the raw materials used for ordinary beef tallow originate from grain-fed beef, which is mainly derived from Henan and Shandong. Therefore, the flavor characteristics and application of these two types of beef tallow are different. Consumer acceptability and market competitiveness are directly affected by the odor of food [7]; therefore, aroma is an important indicator for assessing beef tallow in terms of the product quality of beef tallow hotpot seasoning. However, there is still uncertainty about the quality characteristics and distinguishing methods of the two types of beef tallow. The identification of the complex flavor constituents in food remains a challenge. Over the past 100 years, GC-MS, GC-Q-TOP/MS, GC-Orbitrap-MS, GC-O, GC-IMS, and E-noses have been applied to identify the flavor compounds in various types of food [8,9]. In recent years, in particular, gas chromatograph–ion migration spectrometry (GC-IMS) has been employed to characterize volatile flavor components for food classification and quality control. This method has shown great merits, such as the fact that samples do not need to be pretreated, and benefits in terms of its high sensitivity, fast detection speed, high separation efficiency, and satisfactory visualization [10,11]. Recently, GC × GC-TOF/MS, flash GC electronic nose, GC-MS, and GC-O have been employed to identify the characteristic aromatic components of different kinds of beef tallow and hotpot seasoning products [5,12,13]. In addition, research trends highlight the key differential compounds based on a multivariate statistical analysis (PLS-DA or OPLS-DA), which can be used to better understand the flavor profiles of different food products [14]. However, studies on the identification fingerprinting and differential analysis of the volatile organic components in different types of beef tallow using GC-IMS and an E-nose are rarely published.

In this investigation, we explored the differences in the volatile components of two types of beef tallow by combining an E-nose and GC-IMS. Moreover, the key differential compounds were elucidated through a multivariate statistical analysis.

## 2. Materials and Methods

### 2.1. Materials

The beef tallow samples used in this study were collected from Guanghan Maidele Food Co., Ltd. (manufacturer A) (Chengdu, China) and Sichuan Hangjia Biotechnology Co., Ltd. (manufacturer B) (Chengdu, China). Collect 2–3 batches of samples from each manufacturer, and three samples are randomly selected from each batch. Information about the samples is shown in Table 1.

### 2.2. Electronic Nose Analysis

Five grams of the beef tallow sample was placed into 40 mL headspace injection bottles. The samples were left for 30 min at room temperature to facilitate the emission of the volatile compounds before further processing using the E-nose device. The parameters of the E-nose device (PEN3, Mecklenburg-Vorpommern, Germany) were as follows: a pre-sampling time of 5 s; a zero-point trim time of 5 s; a measurement time of 120 s; a flush time of 150 s; a chamber flow of 300 mL/min; and an initial injection flow of 300 mL/min. The aromatic characteristics of each sample were described via the response values corresponding to the 10 sensors, as presented in Table 2.

### 2.3. HS-GC-IMS Analysis

The VOCs were analyzed through the use of a FlavourSpecr^®^ (Gesellschaft für analytische sensorsysteme GmbH, Dortmund, Germany), which was equipped with an automatic sampler unit (CTC analytics AG, Zwingen, Switzerland). Briefly, 2 g of beef tallow was sampled in a 20 mL headspace vial and then incubated at 60 °C for 15 min. Subsequently, 500 μL of the sample headspace gas was automatically injected into the injector through the use of a heated syringe at 65 °C. Then, the separation of volatile components was performed using GC-IMS, and the GC conditions were as follows: an FS-SE-54-CB-1 capillary column (15 m × 0.53 mm), a film thickness of 1 μm (RESTEK, Bellefonte, PA, USA), a column temperature of 60 °C, carrier gas/drift gas N_2_ (with a purity of 99.999%), and an initial gas flow rate of 2.0 mL/min, which was held for 2 min and then ramped up to 100 mL/min for 2~10 min. IMS conditions were as follows: a temperature of 45 °C and a drift gas flow rate of 150 mL/min. LAV software version 2.2.1 (Gesellschaft fur analytische sensorsysteme GmbH, Dortmund, Germany) was used to determine the analytical spectrum. The volatile compounds were characterized through comparing drift time (DT) and RI through the use of IMS database retrieval software 2.1 (GAS, Dortmund, Germany) and the NIST 11 library. The relative quantitative analysis of volatile compounds was based on the peak intensity in HS-GC-IMS.

### 2.4. Statistical Analysis

The E-nose data were analyzed through the use of Winmuster software (version 1.6.2.18/Sep 25, AIRSENSE Analytics GmbH, Schwerin, Germany), which includes PEN3 E-nose, and a linear discriminant analysis (LDA) was also utilized.

The GC-IMS data were pre-processed using Excel 2016 software and expressed as the mean ± SD (standard deviation), and differences were compared using one-way analysis of variance, with *p* < 0.05 considered as significant and *p* < 0.01 considered as highly significant. Heatmap clustering of VOCs was performed using Origin 2022 (Northampton, MA, USA), and orthogonal partial least-squares discrimination analysis (OPLS-DA) modeling was conducted using SIMCA-14.1 software (Umetrics AB, Umea, Sweden) to quickly and accurately determine the differences in volatile flavor components in the samples.

## 3. Results and Discussion

### 3.1. E-Nose Analysis

A sensor differential contribution rate analysis (loading analysis, LA) is a measure of the magnitude of the sensor’s contribution during the process of discrimination. It can be used to confirm the contribution rate of each sensor to sample differentiation, thus revealing the aromatic components that play the main role in the sample differentiation process.

The results of the loading analysis are shown in Figure 1. The contribution rate of the first principal component (PC-1) was 94.50%, that of the second component (PC-2) was 5.14%, and the total contribution was 99.64%, indicating that the first and second principal components were able to explain 99.40% of the total variance. W1W and W5S showed a high contribution rate on PC-1 and PC-2, indicating that they could be identified as feature sensors.

### 3.2. E-Nose Analysis Combined with LDA

A linear discriminant analysis (LDA) is a commonly used chemometric method that is utilized to reveal the relationship between variables through data dimensionality reduction [15]. As shown in Figure 2, the contribution rate of the first principal component was 67.84%, and that of the second principal component was 4.09%, with a cumulative contribution rate of 71.93%. The first principal component played a major role in the sample, and the greater the distance between two samples on the X-axis, the greater the difference. As shown in Figure 2, the intense flavor beef tallow was mainly distributed between 3.145 and 3.149; in contrast, the ordinary beef tallow was mainly distributed between 3.137 and 3.142. The flavor components of the different types of beef tallow were in regions with an obvious boundary, indicating that the use of an LDA makes it possible to significantly distinguish between the different types of beef tallow.

According to the analysis of different sensor loads, as shown in Figure 1, the difference in the flavors of the two types of beef tallow was mainly related to the information captured by the W5S and W1W sensors, and W5S and W1W were highly correlated with PC1, showing a positive correlation. In turn, W5S (sensitive to nitrogen oxides) and W1W (sensitive to sulfides and terpenes) also showed a high correlation with principal component 2. W1W showed a positive correlation, while W5S showed a negative correlation. Based on the above analysis, it can be concluded that nitrogen oxides and inorganic sulfides can serve as important indicators of differences in odor. Compared with ordinary beef tallow, intense flavor beef tallow has a stronger flavor comprising nitrogen oxides, inorganic sulfides, and terpenes.

### 3.3. GC-IMS Analysis

#### A GC-IMS Spectrum Analysis of the Two Types of Beef Tallow

In order to directly compare the differences in volatile components between the ordinary beef tallow and intense flavor beef tallow, a two-dimensional top view with a blue background was created (Figure 3). The vertical coordinate represents the retention time (s) of the GC, the horizontal coordinate represents the ion migration time (normalized), the red vertical line at the horizontal coordinate 1.0 is the RIP peak (reactive ion peak, normalized), and each spot on both sides of the RIP peak represents a volatile compound [16]. The presence or absence of the spots and the depth of the color indicate the degrees of accumulation and decomposition of the substance, and the color change from white to red is directly proportional to the concentration.

Figure 3 reflects the differences in the types and concentrations of compounds between the ordinary beef tallow and the intense flavor beef tallow. The migration times of both samples were within 1.0–1.5, and the retention time of most compounds was between 100 s and 500 s. The difference in the substance composition was not obvious; however, the intensity of the compound signals in the samples manifested a clear difference. This difference indicated that the variety and concentration of volatile compounds between the two types of samples were different.

### 3.4. The Differences in Volatile Compounds between the Two Types of Beef Tallow

To visually and quantitatively compare the differences in the volatile compounds between the samples, visual fingerprint spectra of the volatile compounds (Figure 4) were established. In Figure 4, each row represents a sample, each column depicts a volatile compound, the color and brightness of each point indicate the compound concentration, “M” and “D” indicate the monomers and dimers of the same compound, and the numbered peaks indicate unidentified peaks.

As shown in Figure 4, the topography of the two samples was relatively similar but with differences in the volatile compound concentration. It was not difficult to establish that most of the components, such as (E)-2-octenal-M(5), (E)-2-heptenal-D, (E)-2-hexenal-D, nonanal-M, nonanal-D, octanal-D, 3-methylbutanal, phenylacetaldehyde, benzaldehyde, gama-butyrolactone, 2-pentanone, 2-hexanone, 2-butanone, 1-octen-3-ol, 1-hexanol, unkonw-3, unkonw-9, unkonw-10, etc., were more abundant in the intense flavor beef tallow. Only a few components, such as hexanal, pentanal, ethyl acetate, and ethanol, were higher in concentration in the normal beef tallow. Differences in the concentration of volatile compounds can bring about changes in flavor. For example, aldehyde compounds can provoke strong citrus, meat, and fat aromas, while alcohol, esters, and ketones can provoke floral and fruit aromas [9,17,18,19], indicating that the flavors of the two types of beef tallow differ significantly.

### 3.5. GC-IMS Integral Parameter Analysis of Volatile Fractions in the Two Types of Beef Tallow

A qualitative analysis of the volatile flavor compounds in the two types of beef tallow was conducted according to the GC retention time and ion migration time. A total of 51 volatile compounds (Table 3) were identified through a comparison with the NIST 2014 gas retention index database and the IMS migration temporal database, including 20 aldehydes, 6 ketones, 5 alcohols, 4 esters, 4 alkenes, 1 acid, 1 sulfur compound, and 10 unknown compounds. Among them, with the exception of the unknown compounds, the most abundant were aldehydes, followed by ketones and alcohols. The proportions of aldehydes, ketones, alcohols, alkenes, esters, acids, sulfur compounds, heterocyclic compounds, and unknown components in the intense flavor beef tallow were 53.1%, 20.0%, 3.1%, 5.1%, 3.9%, 0.2%, 0.2%, 4.4%, and 9.9%, and their proportions were 52.3%, 18.9%, 4.8%, 4.5%, 5.6%, 0.1%, 0.12%, 3.3%, and 10.3% in the ordinary beef tallow (Figure 5). The intense flavor beef tallow was similar to the ordinary beef tallow in terms of the types of volatile compounds; however, their intensities differed significantly. The results of the one-way ANOVA showed that there were significant differences between the two types of beef tallow flavor compounds. With the exception of 11 compounds, including (E)-2-nonenal-D and camphene, which did not show significant differences (*p* > 0.05), the other 40 compounds showed significant or highly significant differences, among which 5 components, including α-terpinene (monomer), showed significant differences (*p* < 0.05), and 35 components, including (E)-2-nonenal, nonenal, etc., showed highly significant differences (*p* < 0.01).

Aldehydes are the most important flavor compounds in beef tallow; they are mainly produced via lipid oxidation and the Maillard reaction [20] and have strong aromatization ability and a low threshold of around 2.5 ppm–0.001 ppm [20,21,22,23], and they mainly present fruit, fat, and nut aromas. As shown in Table 3, the aldehydes in the intense flavor beef tallow and ordinary beef tallow include saturated aldehydes, monoenals, and dieneals; among them, (E)-2-octenal was responsible for the fat and meat aromas; 2-undecenaldehyde and benzaldehyde were responsible for the aldehyde flavor, citrus flavor, fat flavor, and other characteristics; (E)-2-nonanal was responsible for the tallowy, fat, and beefy flavors [24], and the other aldehydes, such as nonanal, phenylacetaldehyde, octanal, heptaldehyde (E)-2-hexenal, hexanal, isovaleraldehyde, etc., were mainly responsible for the fruit and grass aromas [25]. In particular, (E)-2-nonenal and (E)-2-heptenal have been reported as important contributors to beef flavor [26,27]. A single-factor analysis of variance showed that, among the 19 detected aldehydes, 12 aldehydes, including some key aromatic compounds such as (E)-2-nonenal and nonenal, showed significant differences (*p* < 0.05) or highly significant differences (*p* < 0.01), and only 7 aldehydes, namely 2-undecenaldehyde, (E)-2-octenal (dimer), octaldehyde (monomer), octaldehyde (dimer), (E)-2-hepenal, heptanal, and isovaleraldehyde, did not show significant differences. Due to their low odor thresholds and high contents, the differences in these components may lead to flavor differences between the two types of beef tallow.

Ketones are thought to primarily originate from the oxidative dissociation of lipids [28] and have been noted as a major flavor component in beef tallow [5]. Most ketones possess a certain floral and fruity fragrance and have a positive effect on food flavor, and their odor threshold concentration is low [16]. As shown in Table 3, the intensities of all of the ketone peaks were higher in the intense flavor beef tallow, and the hydroxyacetone, γ-butyrolactone, 2-butanone, and 2-heptanone contents were significantly higher in the intense flavor beef tallow.

Alcohols are produced through the oxidative decomposition of oils and fats, and they have a high threshold [5]. As shown in Table 3, 1-octen-3-ol, 1-hexanol, ethanol, and 1-pentanol were the main alcohols found in the beef tallow. The results of the one-way ANOVA showed that, except for ethanol, 1-octen-3-ol, 1-hexanol, and 1-pentanol were significantly more abundant in the intense flavor beef tallow. Straight-chain alcohols, 1-hexanol, and 1-pentanol were the main lipid oxidation products, and they present musty, sweet, and woody flavors and fuel oil, sweet, and balsam odors, respectively [5]. 1-octen-3-ol is a compound that is usually detected in oils or foods that are rich in oils, and its formation often relates to the oxidation of polyunsaturated fatty acid, showing mushroom, rose, and hay scents [29,30]. The results of the one-way ANOVA showed that there was no significant difference in the relative percentage content of this substance in the two types of beef tallow (*p* > 0.05).

Regarding esters, only three compounds were detected (butyl acetate, ethyl acetate, and γ-butyrolactone). They are mainly produced via the esterification of alcohols and free fatty acids during the oxidation of fats and provide a fruity and sweet odor, which is also associated with beef tallow, as shown in previous studies [5,31,32]. The information displayed in Table 3 shows that the content of γ-butyrolactone was significantly higher in the intense flavor beef tallow, and the ethyl acetate content was significantly higher in the ordinary beef tallow. Of note, there was no difference between the two types of beef tallow. γ-butyrolactone is a molecule used as a natural–identical substance in flavoring formulations, which has a faint sweet, aromatic and buttery flavor and has been identified in various natural food matrices and foodstuffs [33]. Ethyl acetate, which is formed from ethanol and acetic acid, contributes a pineapple odor and is a key odor component in beef [12].

One sulfur-containing compound (dimethyl disulfide) and three terpenoids (α-terpinene, camphene, and myrcene) were detected via GC-IMS. Dimethyl disulfide is mainly formed through the thermal degradation of sulfur amino acids. Research has shown that sulfur-containing compounds possess a low flavor threshold and have a significant impact on the overall flavor of beef [34], and the results displayed in Table 3 show that dimethyl disulfide is higher in intense flavor beef tallow. Although there was no significant difference in the content of sulfur compounds between the two types of fat, there is still a possibility of a significant flavor differences due to its low threshold and strong odor. Terpenoids are a more abundant class of compounds; they are mostly produced following the breakdown of lipids, giving beef tallow citrus/herbal and sweet aromas. The content of α-terpinene and myrcene was higher in the intense flavor beef tallow, and some terpenoids such as myrcene possess a low threshold (0.0012 μg/mL) [35,36]. The results suggest that sulfur-containing compounds and terpenoids may play key roles in the flavor profile of beef tallow, which is in agreement with the results of the E-nose analysis.

### 3.6. Similarity Comparison of Volatile Components via Cluster Models

Cluster models were used to analyze the difficult-to-find and complex variables and to distinguish between differences in volatile components in the different samples. All of the GC-IMS spectrum data of the two group samples were analyzed via a cluster comparison, and the results are shown in Figure 6.

The results show that the volatile components of the intense flavor beef tallow were relatively clustered together. These findings further confirmed the differences in the volatile components in the different types of beef tallow. Thus, the volatile compounds of the two types of beef tallow can be suitably distinguished via a cluster analysis.

### 3.7. An OPLS-DA of the Two Types of Beef Tallow

The fingerprint spectra only roughly distinguished the volatile components in the different types of beef tallow, and it was difficult to precisely establish the volatile components that contributed to the differences between the samples. In order to effectively discriminate the flavor differences between the different varieties of beef tallow, two groups of samples were analyzed using multivariate statistical methods, of which OPLS-DA is a supervised statistical analysis method that can be used to model the relationship between compound expression and samples, providing a powerful method for distinguishing between samples with different characteristics [37]. Therefore, the OPLS-DA model was used in this study to identify the specific marker compounds responsible for the aromatic differences in the two types of beef tallow based on the data matrix of the detected aroma-active compounds. As shown in Figure 7, the forecast capacity (*Q*^2^ = 0.971), goodness-of-fit parameter (*R*^2^*X* = 0.74), and explanatory ability (*R*^2^*X* = 0.986) indicate that the findings regarding the flavor compounds within the different types of beef tallow were covered by the fitting equation. According to the results shown in Figure 7, the intense flavor beef tallow was only distributed on the left side of the Y-axis. In contrast, the ordinary beef tallow was distributed on the right side of the Y-axis, and the results are consistent with the fingerprint profiles and heatmap clustering, indicating that there is a significant difference between the intense flavor beef tallow and ordinary beef tallow. In addition, to avoid overfitting, a permutation test (*n* = 200) was performed to evaluate the reliability of the OPLS-DA model, and the results are shown in Figure 7; after 200 cross-validations, all test R^2^ and Q^2^ values were lower than the original values, and the regression line of Q^2^ crossed the abscissa and demonstrated a negative intercept with the vertical axis, indicating that the model was not over-fitted and that it was stable and reliable.

The key compounds responsible for the aromatic profile differences between the two types of beef tallow were further analyzed according to their load values (Figure 8). For example, nonanal, (E)-2-octenal, etc., appeared in higher concentrations in the intense flavor beef tallow compared to the ordinary beef tallow, while ethyl acetate and pentanal occurred in higher concentrations in the ordinary beef tallow. Moreover, the variable importance projection (VIP) was calculated via the construction of a reliable OPLS-DA simulation and used to quantify the impact of each component for classification, and the volatile chemicals with VIP > 1 were considered as the differential marker components in the different samples [38,39,40]. According to the results for the single-variable criterion of the one-way ANOVA (*p* < 0.05) and VIP > 1.0, except for the unknown components, 23 aromatic compounds were screened out as differential marker components in the different samples, including 1-hexanol (fruity aroma), 3-methylbutyric acid, 2,5-dimethylfuran, 2-heptanone (pear-like fruit aroma), 2-hexanone, gamma-butyrolacton, phenylacetaldehyde (sweet aroma), 1-octen-3-ol (mushroom, lavender, rose, and hay aromas), 2-pentaone (acetone-like odor), (E)-2-nonenal-M (fatty green cucumber, citrus, and cardboard flavors), nonanal-D (rose and citrus aromas with a strong oily odor), nonanal-M, (E)-2-octenal-D, (E)-2-octenal-M (fat and meat aromas, with cucumber and chicken aromas as well), (E)-2-heptenal-D (green grass aroma), octanal-D (fruity aroma), ethyl acetate, (E)-2-hexenal-D (fresh green leaf aroma), 3-methylbutanal, 2-undecenal, hexyl 2-methylbutanoate (spicy flavor), 2-butanone (acetone-like odor), and benzaldehyde (bitter almond, cherry, and nut aromas). Only ethyl acetate was dominant in the ordinary beef tallow sample, whereas the other 22 components were dominant in the intense flavor beef tallow sample. The higher content of volatile components makes the flavor of intense flavor beef tallow more typical and richer.

## 4. Conclusions

In this study, an E-nose, in combination with GC-IMS, was employed to investigate the flavors of intense flavor beef tallow and ordinary beef tallow. An E-nose combined with an LDA can be used to distinguish intense flavor beef tallow from ordinary beef tallow, and intense flavor beef tallow is characterized by a stronger flavor comprising nitrogen oxides, inorganic sulfides, and terpenes. A total of 51 volatile compounds were detected in the two types of beef tallow, including 20 aldehydes, 6 ketones, 5 alcohols, 4 esters, 4 alkenes, 1 acid, 1 sulfur compound, and 10 unknown compounds. In terms of the types of volatile compounds, no differences were found; however, the concentrations differed significantly. The differences in volatile compounds in the different types of beef tallow can be satisfactorily recognized through the use of GC-IMS data together with a heatmap clustering analysis and OPLS-DA. A total of 23 differential volatile components were screened out as volatile markers in order to distinguish between the two types of beef tallow. Sulfur-containing compounds and terpenoids might be the key substances that distinguish the flavors of the two types of beef tallow. In summary, based on an E-nose and GC-IMS, we were able to establish the aromatic characteristics and fingerprint spectra of the volatile compounds in the two types of beef tallow examined, and this study might provide a theoretical basis for the quality control and analysis of the characteristics of different types of beef tallow in the future.

## Figures and Tables

**Figure 1 foods-13-01489-f001:**
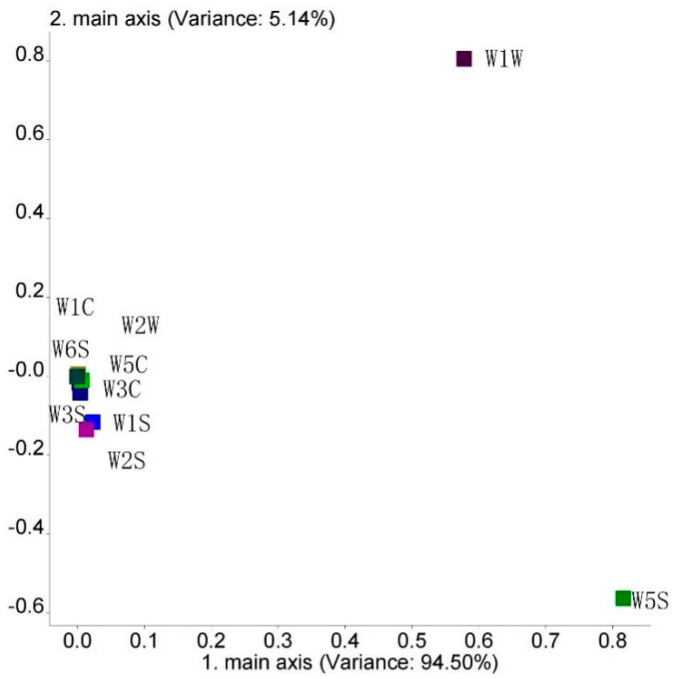
The results of the loading analysis.

**Figure 2 foods-13-01489-f002:**
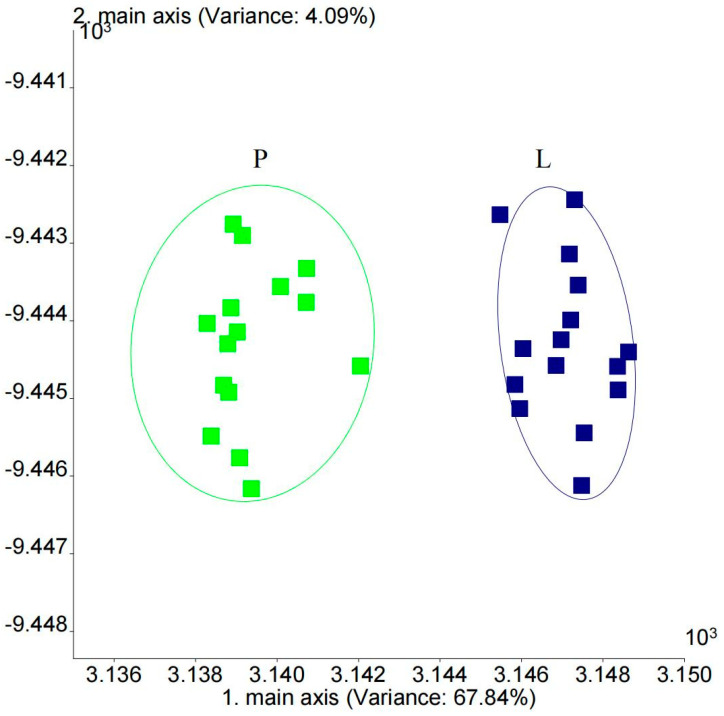
LDA of beef tallow flavor using electronic nose.

**Figure 3 foods-13-01489-f003:**
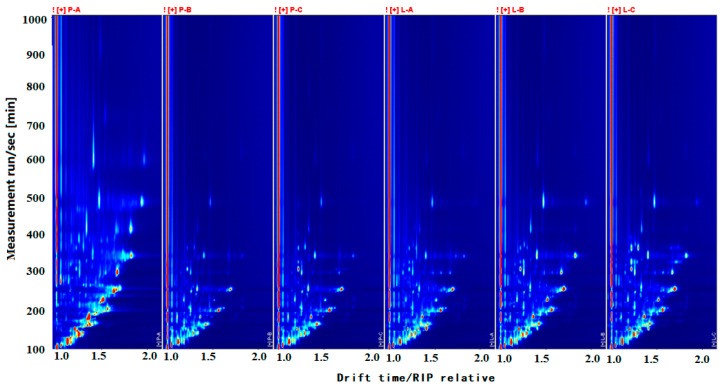
GC-IMS top view of sample spectrum.

**Figure 4 foods-13-01489-f004:**
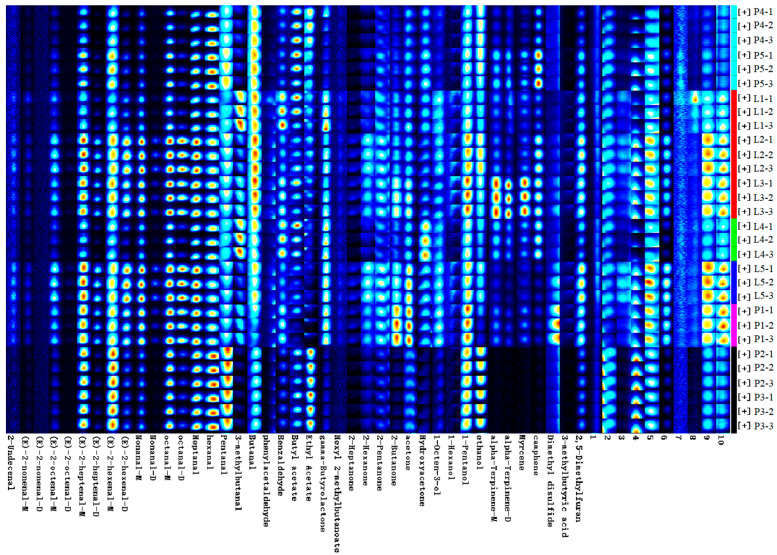
A gallery plot of the two types of beef tallow. P denotes ordinary beef tallow and L denotes intense flavor beef tallow. The letters M and D after the compound indicate the monomer and dimer of said compound.

**Figure 5 foods-13-01489-f005:**
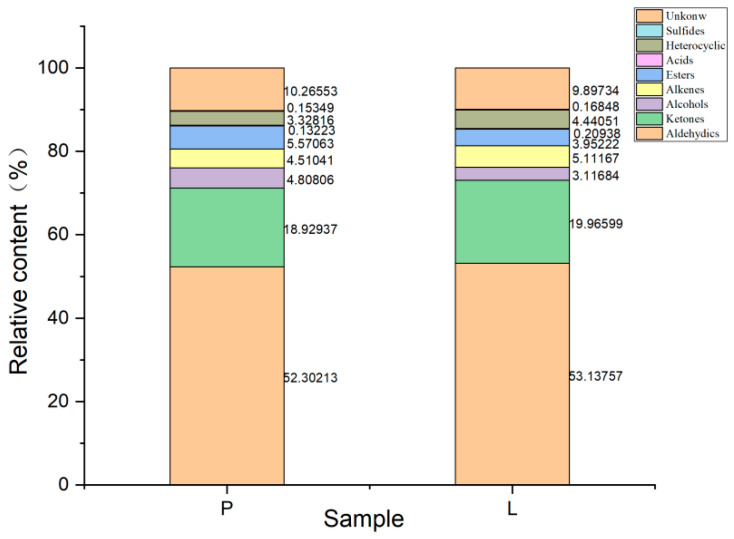
The volatile compounds in the different beef tallow samples determined using GC–IMS.

**Figure 6 foods-13-01489-f006:**
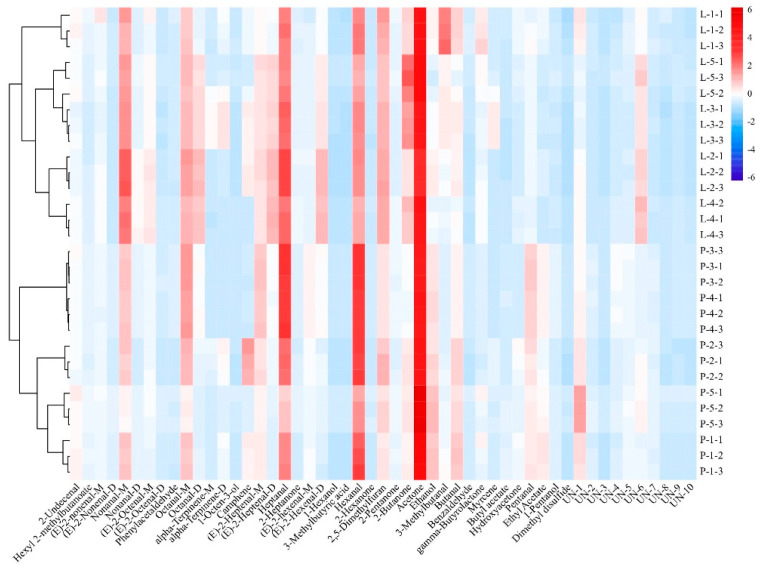
A cluster analysis of the volatile substances in the different types of beef tallow.

**Figure 7 foods-13-01489-f007:**
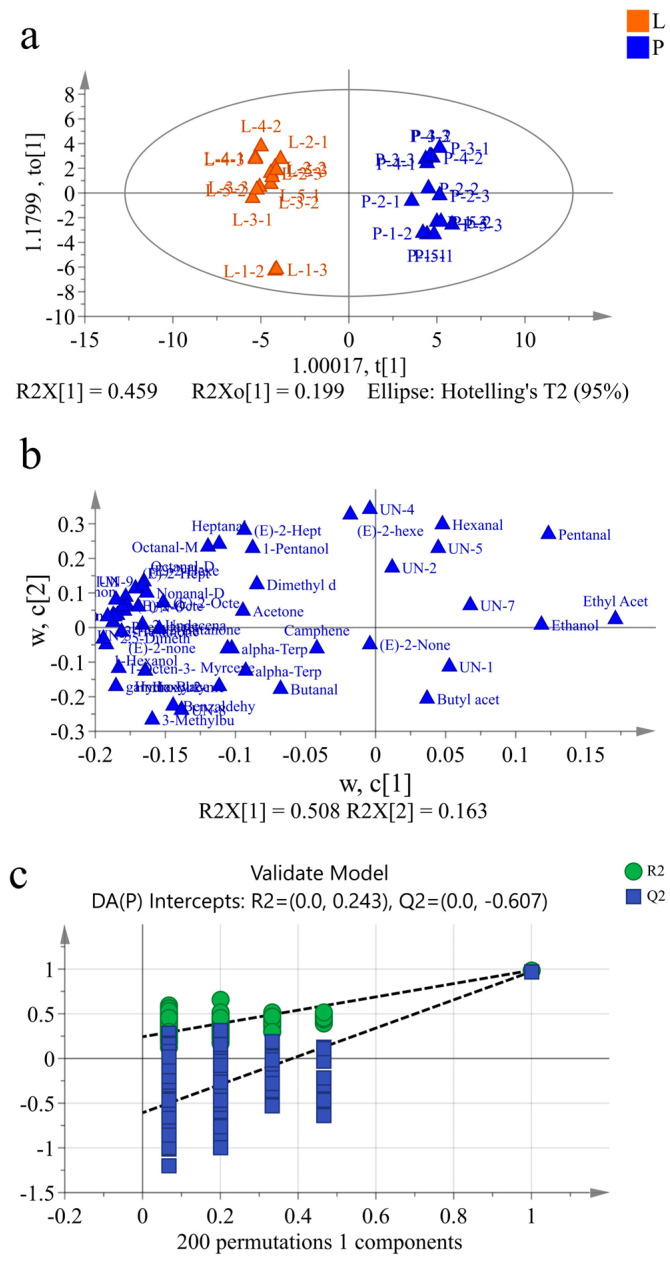
An OPLS-DA (**a**–**c**) of the volatile substances in the different types of beef tallow.

**Figure 8 foods-13-01489-f008:**
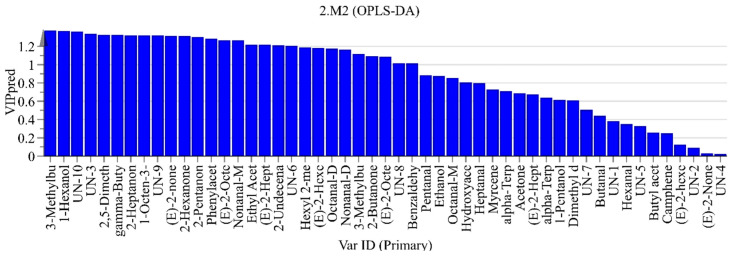
Variable importance projection (VIP) values of OPLS-DA.

**Table 1 foods-13-01489-t001:** Sample information.

Classification	Class	Sample Number	Notes
Intense flavor beef tallow	L	L1	Batch 1 from manufacturer A
		L2	Batch 2 from manufacturer A
		L3	Batch 3 from manufacturer A
		L4	Batch 1 from manufacturer B
		L5	Batch 2 from manufacturer B
Ordinary beef tallow	P	P1	Batch 1 from manufacturer A
		P2	Batch 2 from manufacturer A
		P3	Batch 3 from manufacturer A
		P4	Batch 1 from manufacturer B
		P5	Batch 2 from manufacturer B

**Table 2 foods-13-01489-t002:** The corresponding aromatic types of the different sensors of the electronic nose.

Array No.	Sensor Name	Performance Description
S1	W1W	Sensitive to inorganic sulfides and terpenes
S2	W1S	Sensitive to methyls
S3	W2S	Sensitive to alcohols, aldehydes, and ketones
S4	W3S	Sensitive to long-chain alkanes
S5	W2W	Aromatic ingredients, sensitive to organic sulfur compounds
S6	W1C	Sensitive to aromatic benzene
S7	W6S	Mainly selective to hydrides
S8	W5C	Short-chain alkanes, sensitive to aromatic compounds
S9	W3C	Ammonia, sensitive to aromatic components
S10	W5S	Very sensitive to nitrogen oxides, especially negative to nitrogen oxides

**Table 3 foods-13-01489-t003:** Information on the volatile compounds identified via GC-IMS.

No.	Compound	CAS	RI	Rt [s]	Peak Intensities	
					L	P
Aldehydics	2-Undecenal	C2463776	1400.2	921.264	543.22 ± 63.35 **	408.45 ± 37.76
	(E)-2-Nonenal-M	C18829566	1187.1	614.745	526.96 ± 86.90 **	303.25 ± 32.18
	(E)-2-Nonenal-D	C18829566	1183.5	609.657	179.13 ± 21.44	178.42 ± 13.27
	Nonanal-M	C124196	1107.5	500.277	1760.46 ± 415.91 **	736.39 ± 175.76
	Nonanal-D	C124196	1106.6	499.005	506.46 ± 171.03 **	206.19 ± 27.39
	(E)-2-Octenal-M	C2548870	1055.8	425.967	710.09 ± 178.21 **	299.16 ± 48.24
	(E)-2-Octenal-D	C2548870	1053.8	422.999	189.42 ± 44.64 **	121.96 ± 13.91
	Phenylacetaldehyde	C122781	1036.5	398.26	245.98 ± 37.90 **	127.89 ± 34.13
	Octanal-M	C124130	1005.2	353.235	1296.18 ± 253.06 **	916.82 ± 301.66
	Octanal-D	C124130	1004.2	351.751	968.09 ± 360.33 **	287.34 ± 116.11
	(E)-2-Heptenal-M	C18829555	954.4	305.737	857.48 ± 194.21 *	668.43 ± 192.24
	(E)-2-Heptenal-D	C18829555	953.8	305.242	1037.32 ± 374.15 **	275.36 ± 101.52
	Heptanal	C111717	900.7	261.207	2025.22 ± 363.43 **	1488.67 ± 495.88
	(E)-2-Hexenal-M	C6728263	847.0	231.52	404.12 ± 76.13	388.41 ± 104.00
	(E)-2-Hexenal-D	C6728263	845.5	230.755	965.74 ± 356.60 **	290.55 ± 110.20
	Hexanal	C66251	798.1	206.995	1581.85 ± 142.58	1701.70 ± 344.57
	Pentanal	C110623	688.4	163.264	486.62 ± 54.99	641.02 ± 143.64 **
	3-Methylbutanal	C590863	637.6	151.78	809.91 ± 350.19 **	227.02 ± 76.00
	Butanal	C123728	586.1	140.137	703.51 ± 122.33	630.29 ± 98.25
	Benzaldehyde	C100527	953.3	304.825	152.53 ± 45.34 **	91.90 ± 7.68
Alcohols	Ethanol	C64175	426.1	104.007	545.00 ± 166.72	735.54 ± 112.53 **
	1-Octen-3-ol	C3391864	983.0	329.486	136.97 ± 24.24 **	70.92 ± 9.70
	1-Hexanol	C111273	867.0	241.496	75.03 ± 12.51 **	31.27 ± 4.35
	1-Pentanol	C71410	756.9	189.535	215.96 ± 31.72 *	182.12 ± 43.28
Ketones	2-Hexanone	C591786	777.0	197.23	168.71 ± 28.87 **	89.05 ± 10.14
	Acetone	C67641	493.6	119.246	3408.19 ± 659.32 *	2835.62 ± 485.20
	Hydroxyacetone	C116096	618.6	147.481	324.38 ± 48.92 **	221.05 ± 108.25
	2-Pentanone	C107879	686.1	162.739	399.65 ± 54.64 **	259.32 ± 23.46
	2-Butanone	C78933	570.9	136.712	1346.01 ± 565.07 **	467.27 ± 122.81
	2-Heptanone	C110430	888.5	252.301	414.42 ± 82.78 **	170.69 ± 29.05
Alkenes	α-terpinene-M	C99865	1018.3	372.037	302.43 ± 233.61 **	105.79 ± 98.63
	α-terpinene-D	C99865	1021.8	376.984	421.39 ± 294.30 *	192.74 ± 166.68
	Myrcene	C123353	987.7	333.396	343.52 ± 264.92 **	118.82 ± 90.25
	Camphene	C79925	966.3	315.632	488.92 ± 272.66	355.85 ± 393.66
Esters	Ethyl Acetate	C141786	589.0	140.802	254.22 ± 85.22	465.02 ± 64.16 **
	Butyl acetate	C123864	799.9	207.896	151.10 ± 36.09	162.75 ± 30.24
	γ-Butyrolactone	C96480	916.3	274.181	585.13 ± 97.86 **	234.91 ± 90.13
	Hexyl 2-Methylbutanoate	C10032152	1268.4	731.756	315.31 ± 31.99 **	251.73 ± 16.46
Acid	3-Methylbutyric acid	C503742	832.5	224.246	64.48 ± 9.30 **	26.31 ± 5.68
Sulfides	Dimethyl disulfide	C624920	724.5	177.094	47.63 ± 29.36 *	29.56 ± 5.74
Heterocyclic compounds	2,5-Dimethylfuran	C625865	742.4	183.972	1365.13 ± 200.14 **	677.36 ± 145.30
Unknown	Unknown-1	/	/	/	558.29 ± 99.78	659.91 ± 265.71
	Unknown-2	/	/	/	176.46 ± 20.33	179.13 ± 26.97
	Unknown-3	/	/	/	117.29 ± 27.59 **	36.77 ± 9.39
	Unknown-4	/	/	/	248.48 ± 63.21	245.61 ± 96.58
	Unknown-5	/	/	/	293.58 ± 21.11	304.77 ± 30.28
	Unknown-6	/	/	/	949.31 ± 315.06 **	321.22 ± 94.02
	Unknown-7	/	/	/	203.16 ± 24.41	221.95 ± 31.84
	Unknown-8	/	/	/	130.14 ± 46.72 **	72.09 ± 10.57
	Unknown-9	/	/	/	202.12 ± 37.82 **	86.09 ± 22.16
	Unknown-10	/	/	/	123.72 ± 12.92 **	70.82 ± 10.62

The values are the means ± standard deviation; * and ** represent statistically significant (*p* < 0.05) and highly significant (*p* < 0.01) differences; / represents compounds which could not be identified; and L and P represent intense flavor beef tallow and ordinary beef tallow, respectively.

## Data Availability

The original contributions presented in the study are included in the article, further inquiries can be directed to the corresponding author.

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
