# Peer review of "Insights into the Volatile Flavor Profiles of Two Types of Beef Tallow via Electronic Nose and Gas Chromatography–Ion Mobility Spectrometry Analysis"

_foods, 2024, doi:10.3390/foods13101489_

Round 1
Reviewer 1 Report
Comments and Suggestions for Authors
Line 43 point instead of comma to start a sentence.
Line 61 extra space before GC-MS
Materials and methods
Why you have 15 samples, that means 3 replicates per sample? Could you please add in m&m section? Did you carry out the extraction of three different part of the sample of you analysed three times the same extract? Specify please.
Can you give more information about the samples? Maybe color difference or chemical composition? Fatty acids are related to VOCs profile, maybe this information is also useful to improve the manuscript content.
Figure 4 letters are not legible, please improve the picture quality
Figure 5 Maybe the figure 5a is not necessary if it is in the text…. With the % abundance, should be enough information, maybe you could reorganize the Table 3 following the classification in Figure 5a
Figure 6 letters too small, please modify
Reviewer 2 Report
Comments and Suggestions for Authors
The presented manuscript entitled: ''Insights into the volatile flavor profiles of two types of beef tallow via E-nose and GC-IMS'' describes volatile profiles a beef tallow intense flavor beef tallow and ordinary beef tallow, obtained from two industries. The manuscript is well described but, I have a few comments.
Line 26: from what properties or presence of compounds are ''strong antioxidant properties'' of beef tallow?
Line 31: please rephrase the sentence
Line 72: In how many repetitions was the experiment conducted? How many samples constituted one biological repetition? How many repetitions constituted statistical repetitions of the analysis?
Line 72: on what basis did the authors divide intense flavor beef tallow and ordinary beef tallow?
Line 73: the lack of information about the origin of the samples? what companies do they come from?
Line 73: the lack of legend, in the table
Line 81: Please specify equipment characteristics
Line 142: please change the title of the subsection
Line 219: the table isn't clear, and is not very readable
Line 302: the lack included in the methodology OPLS-DA analysis
Line 354: according to the authors, which is the more reliable method, for analyzing volatile compounds?
The manuscript needs general improvement, it lacks basic elements to repeat the experiment and verify or compare the results obtained. Above all, the description of the trials is missing making it difficult to conduct a fair discussion. As it stands, the article is not suitable for publication in a reputable journal with such a high IF.
Comments on the Quality of English Language
Moderate editing of English language required
Author Response
|
Comments 1: Line 26: from what properties or presence of compounds are ''strong antioxidant properties'' of beef tallow? |
|
Response 1: Thank you for pointing this out. While therre are studies suggest that beef tallow has antioxidant activity due to its presence of vitamin A, vitamin E, and mineral selenium. We have included references in this response[Wang, J., Chen, L., Liu, Y., Olajide, T. M., Jiang, Y. R., & Cao, W. M. (2022). Identification of key aroma-active compounds in beef tallow varieties using flash GC electronic nose and GC× GC-TOF/MS. European Food Research and Technology, 248(7), 1733-1747. ]. If the editor or reviewer does not agree, we will delete the disputed point. |
|
Comments 2: Line 31: please rephrase the sentence |
|
Response 2: Agree. We have rephrase the sentence on line 31. |
|
Comments 3: Line 72: In how many repetitions was the experiment conducted? How many samples constituted one biological repetition? How many repetitions constituted statistical repetitions of the analysis? |
|
Response 3: Thank you for pointing this out, among the 15 samples, there are 3 replicates per sample, and we have supplemented the specific sample information in the m&m section. |
|
Comments 4: Line 72: on what basis did the authors divide intense flavor beef tallow and ordinary beef tallow? |
|
Response 4: Thank you for pointing this out. We agree with this comment. We have described two types of butter in the introduction, these two types of butter are commonly used terms in the industry. They are mainly differ in terms of raw materials, and used for diffrent type of hotpot seasoning. The raw materials used for intense-flavor beef tallow are grass-fed beef which mainly comes from Xinjiang and Inner Mongolia, and raw materials used for ordinary beef tallow are from grain-fed beef which mainly comes from Henan and Shandong. |
|
Comments 5: Line 73: the lack of information about the origin of the samples? what companies do they come from? |
|
Response 5: Thank you for pointing this out. We have added sample sources in the materials section. |
|
Comments 6:Line 73: the lack of legend, in the table |
|
Response 6: Please further specify by the reviewer or editor. |
|
Comments 7:Line 81: Please specify equipment characteristics |
|
Response 7: Please further specify by the reviewer or editor |
|
Comments 8:Line 142: please change the title of the subsection |
|
Response 8: Thank you for pointing this out. We have change the the title in line 142. |
|
Comments 9:Line 219: the table isn't clear, and is not very readable |
|
Response 9: Thank you for pointing this out. According to the first reviewer's comments, we delete the figure 5a,and reorganize the table 3 following the classification in Figure 5a. |
|
Comments 10:Line 302: the lack included in the methodology OPLS-DA analysis |
|
Response 10: Please further specify by the reviewer or editor |
|
Comments 11:Line 354: according to the authors, which is the more reliable method, for analyzing volatile compounds? |
|
Response 11: This investigation focus on the differences in flavor between two types of beef tallow, and whether these differences can be characterized by electronic nose and GC-IMS. The two methods used in this investigation are based on different principles and perspectives to study the differences in butter flavor. The electronic nose mainly imitate human olfaction to analyze whether there are differences in odor, while GC-IMS analyzes the differences in volatile substances. From the results, both methods were able to identify the differences in volatile substances between the two types of butter, and these differences are explained in our manuscript. |
|
4. Response to Comments on the Quality of English Language |
|
Point 1:Moderate editing of English language required |
|
Response 1: In order to improve the quality of english language, we have sought rapid services from MDPI , thus avoiding the negative impact caused by english language errors. |